# Mucosal Plasma Cell Activation and Proximity to Nerve Fibres Are Associated with Glycocalyx Reduction in Diarrhoea-Predominant Irritable Bowel Syndrome: Jejunal Barrier Alterations Underlying Clinical Manifestations

**DOI:** 10.3390/cells11132046

**Published:** 2022-06-28

**Authors:** Cristina Pardo-Camacho, John-Peter Ganda Mall, Cristina Martínez, Marc Pigrau, Elba Expósito, Mercé Albert-Bayo, Elisa Melón-Ardanaz, Adoración Nieto, Bruno Rodiño-Janeiro, Marina Fortea, Danila Guagnozzi, Amanda Rodriguez-Urrutia, Inés de Torres, Ignacio Santos-Briones, Fernando Azpiroz, Beatriz Lobo, Carmen Alonso-Cotoner, Javier Santos, Ana M. González-Castro, Maria Vicario

**Affiliations:** 1Laboratory of Translational Mucosal Immunology, Digestive System Research Unit, Vall d’Hebron Institut de Recerca, Passeig Vall d’Hebron 119-129, 08035 Barcelona, Spain; crispardocamacho@gmail.com (C.P.-C.); john-peter.ganda-mall@oru.se (J.-P.G.M.); elba.exposito@vhir.org (E.E.); merce.albert@vhir.org (M.A.-B.); emelon@clinic.cat (E.M.-A.); marinaforgui@gmail.com (M.F.); dguagnozzi@vhebron.net (D.G.); 2Laboratory of Neuro-Immuno-Gastroenterology, Digestive System Research Unit, Vall d’Hebron Institut de Recerca, Passeig Vall d’Hebron 119-129, 08035 Barcelona, Spain; marc.pigrau@vhir.org (M.P.); nieto@vhebron.net (A.N.); brunokotska@hotmail.com (B.R.-J.); beatriz.lobo@vhir.org (B.L.); carmen.alonso@vhir.org (C.A.-C.); javier.santos@vhir.org (J.S.); 3Facultat de Medicina, Universitat Autònoma de Barcelona, 08193 Bellaterra, Spain; amanda.rodriguez@vhir.org (A.R.-U.); itorres@vhebron.net (I.d.T.); 4Department of Biomedical and Clinical Sciences, Linköping University, 58185 Linköping, Sweden; 5Vascular and Renal Translational Research Group, Lleida Institute for Biomedical Research Dr. Pifarré. Foundation (IRBLleida), Av. Alcalde Rovira Roure 80, 25198 Lleida, Spain; crismartine@gmail.com; 6Department of Gastroenterology, Vall d’Hebron Hospital Universitari, Passeig Vall d’Hebron 119-129, 08035 Barcelona, Spain; fazpiroz@vhebron.net; 7Department of Mental Health, Vall d’Hebron Hospital Universitari, Passeig Vall d’Hebron 119-129, 08035 Barcelona, Spain; 8Centro de Investigación Biomédica en Red de Salud Mental (CIBERSAM), Instituto de Salud Carlos III, 28029 Madrid, Spain; 9Department of Pathology, Hospital Universitari Vall d’Hebrón, Passeig Vall d’Hebron 119-129, 08035 Bar celona, Spain; 10Facultat Ciències de la Salut, Universitat Ramon LLull-Blanquerna, C/Padilla 326, 08025 Barcelona, Spain; piru_santos@hotmail.es; 11Centro de Investigación Biomédica en Red de Enfermedades Hepáticas y Digestivas (CIBEREHD), Instituto de Salud Carlos III, 28029 Madrid, Spain; 12Department of Gastrointestinal Health, Nestlé Institute of Health Sciences, Société des Produits Nestlé S.A., Nestlé Research, Vers-chez-les-Blanc, 1000 Lausanne, Switzerland

**Keywords:** intestinal plasma cells, intestinal glycocalyx, IBS-D, mucosal ultrastructure, intestinal barrier dysfunction, mucosal nerve fibres

## Abstract

Irritable bowel syndrome (IBS) is a disorder of brain-gut interaction characterised by abdominal pain and changes in bowel habits. In the diarrhoea subtype (IBS-D), altered epithelial barrier and mucosal immune activation are associated with clinical manifestations. We aimed to further evaluate plasma cells and epithelial integrity to gain understanding of IBS-D pathophysiology. One mucosal jejunal biopsy and one stool sample were obtained from healthy controls and IBS-D patients. Gastrointestinal symptoms, stress, and depression scores were recorded. In the jejunal mucosa, RNAseq and gene set enrichment analyses were performed. A morphometric analysis by electron microscopy quantified plasma cell activation and proximity to enteric nerves and glycocalyx thickness. Immunoglobulins concentration was assessed in the stool. IBS-D patients showed differential expression of humoral pathways compared to controls. Activation and proximity of plasma cells to nerves and IgG concentration were also higher in IBS-D. Glycocalyx thickness was lower in IBS-D compared to controls, and this reduction correlated with plasma cell activation, proximity to nerves, and clinical symptoms. These results support humoral activity and loss of epithelial integrity as important contributors to gut dysfunction and clinical manifestations in IBS-D. Additional studies are needed to identify the triggers of these alterations to better define IBS-D pathophysiology.

## 1. Introduction

Irritable bowel syndrome (IBS) is one of the most prevalent gastrointestinal disorders in our society [1], characterised by recurring abdominal pain, discomfort, and altered bowel habits. IBS aetiology and pathophysiology are still not well understood, but a number of factors are associated with the generation of gastrointestinal symptoms, including dysregulation of the hypothalamic-pituitary-adrenal axis, neuroendocrine alterations, immune activation, and visceral hypersensitivity [2,3]. In addition, previous gastrointestinal infection, female sex, and anxiety and depression are identified as risk factors for developing IBS [4,5], but the specific mechanisms leading to the onset of IBS remain unclear.

Although there are no structural or biochemical disease markers for IBS, altered intestinal barrier function, though not universal, has been identified in all subtypes, more consistently in the diarrhoea-predominant IBS subtype (IBS-D) [6,7], in which functional and structural epithelial abnormalities are associated with more severe symptoms [8,9]. Epithelial dysfunction facilitates luminal antigens to access the intestinal mucosa, triggering local and/or systemic inflammatory responses [10]. The gastrointestinal tract is the largest lymphoid organ of the human body and contains diverse populations of plasma cells, T and B lymphocytes, macrophages, dendritic cells, eosinophils, and mast cells. In fact, low-grade inflammation is recognized along with the different intestinal segments in IBS; more specifically, a higher number of mast cells, lymphocytes, and plasma cells [11,12,13], with mast cells being the main immune cell type associated with visceral hypersensitivity, intestinal dysmotility, and epithelial barrier dysfunction. We have previously identified humoral activity as significant in the intestinal mucosa in IBS-D [13] in association with major clinical manifestations.

Moreover, faecal IgA-coated bacteria [14], as well as circulating antibodies against flagellin [15], are found at higher concentrations in IBS-D than in healthy controls. This increase in acquired immunity may respond to incomplete physical and/or biochemical defensive mechanisms at the luminal interface. In IBS, reduced intestinal epithelial integrity has been evidenced as altered expression, phosphorylation, and localization of cell adhesion proteins (mainly tight junction proteins), cytoskeleton condensation, and increased intercellular space [7,8,16,17], presumably underlying the increase in epithelial permeability [18]. The contribution of the mucus layer to barrier defence in IBS remains unknown, despite its fundamental role in preventing direct contact of luminal bacteria with the epithelium [19]. Of special interest is the glycocalyx, a layer of transmembrane mucins at the apical surface that acts as a barrier between secreted mucins and the epithelium and influences several cell signalling pathways, modulates inflammatory responses, and regulates proliferation, differentiation, and apoptosis [20]. In preclinical studies of gut injury and postnatal development, a deteriorated glycocalyx layer has been linked to increased permeability and adherence of microbes to the epithelium [21,22,23]. However, in IBS, the contribution of intestinal glycocalyx to barrier integrity and its potential association with mucosal immunity has not been fully defined.

In this study, we hypothesized that the decreased intestinal epithelial integrity at the luminal interface in IBS-D is linked to increased local humoral activity, likely facilitating mucosal exposure to luminal antigens. We aimed to get further insight into the role of the glycocalyx and plasma cells in mucosal barrier function and its potential contribution to IBS-D pathophysiology.

## 2. Materials and Methods

### 2.1. Participants and Clinical Assessment

Newly diagnosed IBS-D patients fulfilling the Rome III criteria [24] were prospectively recruited from the outpatient gastroenterology clinic of Vall d’Hebrón Hospital Universitari (Barcelona, Spain), and healthy control (HC) subjects were recruited from the general population by public advertising (September 2018 through December 2020). A complete clinical assessment was performed on all participants, including gastrointestinal, psychological, and food allergy-associated symptoms (inclusion and exclusion criteria are summarized in Appendix A). Past episodes of infectious gastroenteritis and other gastrointestinal comorbidities were identified by clinical history assessment and using biochemical and serological tests, including anti-transglutaminase antibodies and thyroid hormones. Participants reported digestive and psychological symptoms daily for ten days previous to the collection of one jejunal mucosal biopsy. Only IBS-D participants completed the IBS severity score system questionnaire (IBS-SSS) [25] to evaluate abdominal pain and distention. They measured the impact on the quality of life and satisfaction with bowel habits symptoms. In all subjects, pain frequency (number of days with pain) over the last days was recorded [25], and to assess bowel habits, stool form (by the Bristol Stool Chart score) and frequency (maximum number of bowel movements) over the last ten days were recorded [25,26]. Depression traits and psychosocial chronic and acute stress were recorded using the validated Spanish versions of the Beck’s Depression Inventory [27], the Modified Social Readjustment Scale of Holmes-Rahe [28], and the Perceived Stress Scale of Cohen [29], respectively. Evaluation of allergy included the clinical history of food-related symptoms (digestive and extra-digestive) and a battery of skin prick tests for 22 common food allergens and 12 inhalants (Laboratorios Leti SA, Barcelona). The study protocol was approved by the Ethics Committee at Vall d’Hebrón Barcelona Hospital Campus (Barcelona, Spain) [(PR(AG)211/2018)], and written informed consent was obtained from all participants. The study was carried out in accordance with the Declaration of Helsinki and the principles of good clinical practice.

### 2.2. Experimental Design and Procedures

Clinical assessment was obtained in all participants and biological samples (stool and a jejunal mucosal biopsy) in specific subgroups. The stool was processed for the quantification of immunoglobulins by ELISA. The jejunal biopsies were split into two similar pieces: one fragment was fixed in 4% buffered formalin and embedded in paraffin for further microscopical examination (histology, immunohistochemistry, and immunofluorescence), and the other fragment was immediately fixed in a fixative buffer for ultrastructure analysis by transmission electron microscopy (TEM) or immersed in RNA Later Solution and stored at −80 °C until further processed for RNA isolation, RNAseq or qPCR analysis. Samples for each experimental procedure were coded and processed blindly. Results were analysed, and the association between biological findings and clinical symptoms was investigated to get further insight into IBS-D pathophysiology.

### 2.3. Collection of Biological Samples

*Stool:* One sample was collected per participant 1 or 2 days prior to the biopsy procedure. The stool was kept at −20 °C, transported to the laboratory at 4 °C, and stored at −80 °C until further processing and analysis.

*Jejunal biopsy*: After an overnight fast, participants were orally intubated, and Watson’s capsule was positioned 10 cm distal to the angle of Treitz under fluoroscopic control to obtain one mucosal biopsy as previously described [30].

### 2.4. Analytical Procedures

#### 2.4.1. Gene Expression Analysis by RNAseq and Q-RT-PCR

*RNA extraction*: Biopsies were lysed in 1 mL of TRIzol Reagent (ThermoFisher Scientific, Carlsbad, CA, USA) using FastPrep (MP Biomedicals, Ohio, USA). Sample processing was performed according to the manufacturer’s instructions. Quantity and quality of total RNA, obtained from the aqueous phase after being filtered and eluted through miRVANA columns, were analysed by capillary electrophoresis (Agilent 2100 Bioanalyzer; Agilent Technologies, Waldbronn, Germany). Only samples with values of RIN ≥ 7 were included in the analysis, from which 20 subjects from each group were blindly selected for RNAseq analysis.

*RNAseq analysis*: One microgram of total RNA was used to create a polyA+ stranded barcoded cDNA library using standard protocols, and samples were sequenced in the Ilumina HiSeq 2500, as previously described [31]. Data were analysed by the Bioinformatics Unit from Bellvitge Biomedical Research Institute (Bellvitge, Spain) using the DESeq2 Bioconductor package (v1.24.0, Heidelberg, Germany) [32]. Gene set enrichment analysis (GSEA) was performed to assess the functional relevance of RNAseq data. Raw counts were normalized by applying the Variance Stabilizing Transformation (VST), and genes were mapped using Ensembl identifiers. The matrix was constructed based on a Gene Cluster Text (GCT) file based on the normalized counts and a Categorical Class (CLS) file based on the metadata that defines IBS-D and HC phenotypes. GSEA was run using Gene Matrix Transposed (GMT) files associated with curated gene sets from the Reactome (C2) and from immunologic signatures (C7) downloaded from the collections of the Molecular Signatures Database (MSigDB v7.0, Broad Institute, Cambridge, USA) [33], together with the CHIP file with the annotation matching Ensemble IDs. The ranking metric used was the signal-to-noise ratio and gene set size filter set to a minimum of 15 and a maximum of 500 genes. Enrichment was considered significant when the FDR was <25%.

*cDNA synthesis and Q-RT-PCR*: cDNA synthesis was performed using 1 µg of total RNA with the High Capacity Reverse Transcription Reagents Kit (Applied Biosystems, Thermo-Fisher Scientific, Vilnius, Lithuania). Gene expression was assessed for MUC17, a transmembrane mucin and a major component of the glycocalyx using validated TaqMan Gene Expression Assays (GEA), and the human 18S subunit ribosomal RNA gene as the endogenous control (Applied Biosystems), by Q-*RT*-PCR on an ABI PRISM^®^ 7500 FAST Sequence Detection System (Applied Biosystems) and analysed by the 2^−ΔΔ*C*t^ method, as previously described [13].

#### 2.4.2. Histological Assessment

All tissue specimens were cut at 4 µm, placed onto slides, deparaffinized, and rehydrated following general procedures. Tissue sections were processed for routine hematoxylin and eosin following a general protocol to assess the mucosal morphology and the eosinophilic infiltration. Additionally, the number of intraepithelial lymphocytes and mast cells were identified using mouse anti-human CD3 and mouse anti-human CD117 antibodies, respectively, as previously described [30]. In the jejunum, different from the duodenum, the normal range of intraepithelial lymphocytes per 100 villous epithelial cells has been established as 6–40 [34]. All samples were examined by an experienced pathologist. To identify the proximity between plasma cells and nerve fibres in the jejunal mucosa, double immunofluorescence staining was performed by localization of CD138 (plasma cell marker) and PGP9.5 (neuronal marker) with monoclonal antibodies (Appendix A). Briefly, tissue sections were deparaffinized and dehydrated following general procedures. Antigen retrieval was performed (Tris-EDTA buffer pH 9, 10 min at 120 °C in) followed by permeabilization (0.1% Triton X-100, 5 min at RT) and blocking with Blocking Solution (Dako, Santa Clara, CA, USA). Samples were incubated simultaneously first with the primary antibodies (4 °C, overnight) and secondly with the secondary antibodies (30 min at RT; Appendix A). Nuclei were counterstained with 4′,6-diamidino-2-phenylindole (DAPI). The negative control slides followed the same procedure, excluding the addition of the primary antibodies. Fading was controlled using the Prolong anti-fade mounting media (Molecular Probes). Stainings were visualized using a Zen 3.0 (Blue version, Jena, Germany) and a built-in camera in a confocal fluorescence microscope LSM980 (Zeiss, Jena, Germany).

#### 2.4.3. Ultrastructural Analysis

After fixation in 0.1 M phosphate buffer containing 2.5% (*v*/*v*) glutaraldehyde (Merck, Darmstadt, Germany) and 2% (*w*/*v*) paraformaldehyde (Sigma, Saint Louis, MO, USA) for 48 h at 4 °C, samples were immersed in 1% osmium tetroxide (Sigma) (*w*/*v*) and dehydrated sequentially, following standard procedures. Samples were embedded in Eponate 12 resin (Ted Pella Inc., Redding, CA, USA), polymerized, and ultrathin (70–90 nm) sections were placed on gold grids (100 mesh) with a film layer. Samples were visualized using a TEM JEM-1400 (Jeol Ltd., Tokyo, Japan) equipped with a Gatan Ultrascan ES1000 CCD camera and the Digital Micrograph© software for image obtention and analysis. Only cuts containing both intact epithelium and *lamina propria* were evaluated. A minimum of 20 non-overlapping fields per subject were observed at ×4,000–20,000 magnification (for the assessment of plasma cells) and ×60,000–80,000 magnification (for the assessment of microvilli). Morphometric analysis was performed blindly to evaluate:-Plasma cell activation: The rough endoplasmic reticulum (RER) and the mitochondria were quantitatively assessed as previously described [35] with modifications in 3–4 plasma cells per subject. Results are expressed as RER area (total [µm^2^] and %RER in the cytoplasm) and average number and area (µm^2^) of mitochondria per plasma cell. A detailed description of the process and validation of the method can be found in the Appendix A.-Plasma cell proximity to nerve fibres: The distance between plasma cells and nerves (membrane to membrane) was measured (4–6 measurements per plasma cell) in 7–10 plasma cells per subject. Results are expressed as the average distance in µm.-Glycocalyx thickness: The thickness of the glycocalyx overlying the microvilli of the enterocytes was measured in ten non-overlapping fields per subject. Representative measurements (3–7 per field) that included the highest and lowest thickness were taken, and the results are expressed as the average thickness in nm.

#### 2.4.4. Immunoglobulin Quantification

Immunoglobulins were quantified in the stool. Samples were aliquoted in Lysing Matrix E Tubes (MP Biomedicals, Solon, OH, USA) and diluted in PBS supplemented with Halt^TM^ protease inhibitor cocktail (ThermoFisher, Waltham, MA, USA) to reach a concentration optimal for detection by ELISA (experimentally set), according to the Ig isotype: 0.08 mg/mL for sIgA, 4 mg/mL for IgM, 20 mg/mL for IgG, 100 mg/mL for IgG subtypes and 200 mg/mL for IgE. Samples were homogenized using Mini-Beater-16 cell disrupter (Biospec products, Bartlesville, OK, USA) and centrifuged (4 °C 13,000× *g* 5 min). Total protein content was performed by the Pierce BCA protein assay (ThermoFisher) method, and immunoglobulin concentration was assessed using ELISA kits (Appendix A). Results are expressed in ng/mg of protein and µg/mg protein for sIgA.

### 2.5. Statistical Analysis

Normality of the data distribution was tested by the D’Agostino and Pearson omnibus normality test. For normally distributed parametric data, results are expressed as mean ± SD and compared by the unpaired Student’s *t*-test (two-tailed), while correlations were tested by Pearson’s correlation coefficient. For non-parametric data, results are expressed as the median and the minimum-maximum range and analysed by the Mann-Whitney U test. Fischer’s exact test was used for assessing statistical differences between proportions. Relationships between clinical variables and biological data were assessed by Spearman’s rho correlation. Values of *p* ≤ 0.05 were considered significant. To assess the correlation between clinical and biological variables, the Benjamini–Hochberg procedure was used to test for multiple comparisons, and a false discovery rate (FDR) of ≤25% was considered significant. Analysis was performed using GraphPad Prism 6.0 software (Graphpad Software, San Diego, CA, USA).

## 3. Results

### 3.1. Study Population

A total of 63 HC and 71 IBS-D patients were included in the study (Table 1). The female:male ratio and the median age were higher in the IBS-D group than in the HC group. IBS-D patients scored higher in depression symptoms (Beck’s Inventory) and perceived acute stress (Cohen scale) than HC subjects. In the IBS-D group, 63.4% had dyspepsia, showing significantly higher IBS-SSS scores than non-dyspeptic patients with IBS-D. Stratification by sex in the IBS-D cohort revealed a higher proportion of women (71.4%) with dyspepsia compared to men (26.4%) (*p* = 0.01), and women had more severe symptoms than men (women: 288 (80.0–459); men: 221 (110–384), *p* = 0.008), as well as higher perceived acute stress and depression symptoms (Appendix A). Other clinical characteristics of the participants are shown in Table 1.

#### Jejunal Histology and Immune Cell Counts

Tissue samples contained only the mucosal layer. Normal epithelial architecture and no relevant lymphoplasmacytic infiltrate in the *lamina propria* or excess intraepithelial lymphocytes (all showed ≤ 40 per 100 enterocytes) were detected under routine histological evaluation. In addition, no parasites, microbial, or viral inclusions were observed. Specific staining revealed no significant differences in mucosal mast cell and eosinophil counts between the two groups (Table 1). Stratification by sex in the IBS-D cohort showed that women had a lower number of mast cells and eosinophil in the *lamina propria* (Appendix A).

### 3.2. The Jejunal Mucosa Shows Higher Representation of Humoral Activity Pathways in IBS-D Compared to Health

GSEA analysis of RNAseq data showed 653 gene sets from the Reactome collection upregulated in IBS-D; from those, 208 gene sets were significant (FDR < 25%). Among the top enriched Reactome pathways (Table 2), we found those related to inflammatory and immune responses (IL1 and NF-KB signalling, B-cell receptor activation events, antigen presentation) and epithelial barrier-related pathways (actin dynamics, GAP, and tight junctions activity).

GSEA analysis using gene sets from the immunologic signatures collection rendered 3293 upregulated gene sets in IBS-D; from those, 247 were significant (FDR < 25%). Down and upregulated genes belonging to immunological signatures related to plasma cell activity, germinal centre activity, and predominance of B cells were overrepresented in IBS-D (Table 3).

Among the top 50 genes with positive enrichment scores (at the top of the ranked list) in IBS-D patients (Figure 1A and Appendix A), we observed that 58% were directly related to Ig structure (Figure 1B). From those, the majority (93%) encode for the variable antibody region (67% encodes for light chains and 33% for heavy chains). Of the two types of light chains (kappa and lambda), a higher number of genes encoding for lambda chains were represented compared to kappa light chains (67% and 33%, respectively) and presented higher enrichment scores in the IBS-D group. The Ig isotype is determined by the constant region of heavy chains; interestingly, genes determining IgG2 and IgG3 isotypes were among the top 50 ranked genes in IBS-D (Figure 1B).

### 3.3. Ultrastructural Assessment Shows Differential Plasma Cell Activation in the Jejunal Mucosa in IBS-D

The morphometric quantification of plasma cells images captured by TEM showed a higher area of both the RER (IBS-D: 12.3 (5.1–17.6) µm^2^; HC: 4.6 (2.2–5.8) µm^2^, *p* < 0.001) and the cytoplasm (IBS-D: 30.7 (18.6–55.6) µm^2^; HC: 24.5 (9.8–26.4) µm^2^, *p* = 0.038) in the IBS-D group compared to the HC group (Appendix A). The percentage of RER area in the cytoplasm (%RER), a more representative value of the area of 2D images, was also higher in the IBS-D group than in the HC group (IBS-D: 32.8 (25.3–40.4)%; HC: 18.7 (14.9–25.2)%, *p* < 0.001) (Figure 2B). The average number of mitochondria was also higher in the plasma cells of IBS-D patients compared to HC subjects (IBS-D: 9.3 (3.5–12.7) mitochondria/plasma cell; HC: 4.5 (2.0–7.0) mitochondria/plasma cell, *p* = 0.003) (Appendix A) while the area of the mitochondria did not show any significant differences between the two groups (data not shown). The number of mitochondria and the %RER in cytoplasm positively correlated (*r* = 0.61, *p* = 0.004, pooled participants n = 20, Figure 2C); this may reflect that increased immunoglobulin synthesis in plasma cells implies additional energy requirement.

### 3.4. Reduced Glycocalyx in IBS-D in Association with Plasma Cell Activation

Quantitative analysis showed that the thickness of the glycocalyx in jejunal enterocytes was lower in the IBS-D than in the HC group (IBS-D: 100.5 (79.0–120.0) nm; HC: 159.0 (135.0–200.0) nm; *p* < 0.001; Figure 2D,E). Glycocalyx thickness is inversely correlated with all assessments indicative of plasma cell activation (Appendix A). Notably, the glycocalyx thickness inversely correlated with the %RER in plasma cells (*r* = −0.77, *p* < 0.001, pooled participants n = 16; Figure 2F), suggesting that reduced epithelial barrier properties underlie higher plasma cell activation. We further analysed the gene expression of MUC17, one of the fundamental proteins in forming the glycocalyx meshwork [22], but did not detect any significant difference between IBS-D and HC (Appendix A).

### 3.5. Increased Mucosal Plasma Cell Proximity to Nerve Fibres in IBS-D, in Association with Plasma cell Activation and Glycocalyx Reduction

The assessment of structural markers of plasma cells (CD138) and nerves (PGP9.5) allowed us to identify the proximity between the two cell types within the jejunal mucosa in both groups (Figure 3A). No difference was observed between the two groups (data not shown). However, the quantification of this distance by TEM analysis (Figure 3B) revealed that the distance from plasma cells to nerve fibres was lower in the IBS-D group as compared to the HC group (IBS-D: 0.87 (0.21–4.25) µm; HC: 1.94 (0.26–7.06) µm; *p* < 0.001) (Figure 3C). Notably, this distance inversely correlated with the degree of plasma cell activation, as measured by the %RER in the cytoplasm of plasma cells (*r* = −0.54, *p* = 0.015, pooled participants n = 20; Figure 3D) and positively correlated with the thickness of the glycocalyx (*r* = 0.83, *p* < 0.001, pooled participants n = 16; Figure 3E), suggesting a potential epithelial-neuro-immune axis as relevant in mucosal barrier mechanisms in IBS-D.

### 3.6. Higher Concentration of IgG in the Intestinal Lumen in IBS-D

Quantifying Igs in stool samples revealed a higher concentration of total IgG and the IgG2 subtype in IBS-D compared to HC. Differences observed in sIgA and the IgG3 subtype did not reach statistical significance, and no differences were found in IgM, IgE, IgG1, and IgG4 between groups (Table 4).

### 3.7. Associations between Clinical, Histological, and Biological Variables

To get further insight into the potential relevance of these findings to IBS-D pathophysiology, we evaluated the association between the different variables analysed (Table 5). The number of bowel movements and the stool consistency correlated with both reduced glycocalyx thickness and plasma cell activation. Perceived stress (Cohen’s scale) and depression symptoms also correlated with reduced glycocalyx thickness. The number of mast cells and T cells were inversely associated with glycocalyx thickness, and the number of T lymphocytes correlated with plasma cell activation. Additionally, the amount of IgG in the intestinal lumen, as measured in stool samples, correlated with the intensity (*r* = 0.510, *p* = 0.026; FDR = 0.257) and frequency (*r* = 0.490, *p* = 0.030; FDR = 0.148) of the abdominal pain. Stratification by sex revealed women to show significant inverse correlations between plasma cell-to-nerves distance with both, perceived stress (*r* = −0.61, *p* = 0.002; FDR = 0.026) and depression symptoms (*r* = −0.57, *p* = 0.005; FDR = 0.033), associations that were not observed in men.

## 4. Discussion

The present study confirms the differential humoral activity in the jejunal mucosa as a feature in IBS-D and identifies, for the first time, the reduction of the jejunal glycocalyx and its association with mucosal plasma cell activation and proximity to nerves. This interaction suggests that epithelial barrier dysfunction facilitates the passage of luminal antigens, which may trigger local humoral responses. In fact, epithelial barrier dysfunction is proposed as a central mechanism in IBS pathophysiology. However, its aetiology and the specific mechanisms leading to gut dysfunction are still unknown.

We and others had previously demonstrated that altered intestinal epithelial barrier function is a hallmark of IBS and is linked to mucosal mast cell activation [8,16,36]. After, we identified B lymphocytes and plasma cells as significant contributors to the low-grade mucosal inflammation in association with major clinical manifestations [13]. Now, in an additional IBS-D cohort, we extend our previous findings on epithelial integrity and humoral immunity to identifying reactome pathways associated with epithelial junctional complex and immunological signatures. We report the thinning of the glycocalyx layer overlying the microvilli and increased plasma cell activation and proximity to nerves in IBS-D with respect to health. Notably, the thinning of the glycocalyx is associated with mucosal plasma cell activity, in addition to clinical severity, more specifically with stool pattern and psychological stress and depression. These new findings further add to the scientific evidence of epithelial barrier dysfunction as a central event in IBS and bring new avenues for the potential development of diagnosis and/or prognosis biomarkers.

At the luminal interface, the mucus layer protects the host from the commensal microbiota and invading pathogens. The glycocalyx, a carbohydrate-rich extracellular layer at the surface of the epithelial cells, acts as a size-selective diffusion barrier, preventing the adherence of microorganisms and small molecules to the epithelium [37]. Impaired goblet cell function and dysregulated mucin biosynthesis have been described in IBD, colon cancer, and gastrointestinal infections [20,38]. Interestingly, damage to the colonic mucus layer has been proposed as a shared cause of IBD and IBS [39]. In IBS, upregulation of the mucin-forming MUC20 gene [40] and reduction of the mucus layer and glycocalyx components have been observed in the colon [41]. This later study identifies decreased barrier mechanisms in *Brachyspira* infected patients and not in patients without infection. On the contrary, in our study, we describe a significant reduction of the glycocalyx in the jejunum of all IBS-D patients, in which previous gastrointestinal infections were excluded by anamnesis. We have not studied whether this thinning of the glycocalyx is the consequence of its degradation or synthesis alterations and if the loss of glycocalyx impacts transcellular or paracellular permeability, but the differences found and the strong biological and clinical associations identified in our study position the glycocalyx as a potential contributor to IBS pathophysiology. Serotonin is a neurotransmitter found to decrease in the jejunum of IBS patients [42]. Drugs that increase the availability of serotonin have shown positive results in IBS patients [43], partially perhaps due to increased mucus secretion [44], although the effect of serotonin on glycocalyx is not known to our knowledge.

Of interest, degradation of the glycocalyx could be driven by luminal bacteria. Enteric pathogens, such as *Salmonella Typhimurium*, possess hydrolases that can cleave glycans of the glycocalyx and breach through to the epithelium [45]. We have previously shown this pathogen to translocate at a higher rate across the colonic epithelium in IBS patients than in healthy controls [46], a finding that could be explained by our new findings of reduced glycocalyx thickness. Notably, many members of the commensal microbiota share features that allow them to cleave glycocalyx components, as shown in a pre-print by Martino et al. [47]. Speculatively, this could suggest that frequent conditions in IBS, such as small intestinal dysbiosis or small intestinal bacterial overgrowth [48,49], with a potentially higher number of glycocalyx-degrading microbes, could contribute to the breakdown of the transmembrane mucins layer in IBS, as recently suggested [41], and relate to clinical manifestations. However, additional studies are needed to confirm this new hypothesis.

Stress is known to exacerbate IBS clinical manifestations [50] and to drive IBS-like symptoms and intestinal pathobiological alterations in animal models [51,52]. The effect of such stress on the intestinal glycocalyx is, to our knowledge, unknown in humans; however, it has been shown that O-glycosylation of mucins was strongly affected by water-avoidance stress in rats with flattening and loss of the mucus layer cohesive properties [53]. Of interest, degradation of the glycocalyx could also be driven by drugs such as indomethacin [21], dietary components such as emulsifiers [54], or dietary deprivation of short-chain fatty acids [55].

Our observations also suggest a brain-to-gut driven mechanism based on the clear association between stress and depression scores and the reduction in the transmembrane mucins layer. Changes in glycocalyx composition could also be expected, as RNAseq analysis of mucosal biopsies from ulcerative colitis patients in remission show an increased expression of MUC17 compared to those in active state and controls [56]. MUC17 is highly expressed at the tip of the microvilli in the human ileum [22] and, together with MUC13, plays a pivotal role in forming the meshwork of the glycocalyx. Our results show no difference in MUC17 gene expression between groups, thus not mimicking other findings. However, we cannot exclude other mucus-forming proteins to be differentially expressed or post-transcriptional regulation to also modulate glycocalyx composition. Moreover, other components such as lipids, carbohydrates, and specific amino acids that create binding sites for O-linked oligosaccharides should be analysed to well characterise the glycocalyx in IBS [57]. The functional consequences of this reduction are unknown in IBS, but both in vitro and animal studies demonstrate a deteriorated glycocalyx barrier resulting in increased permeability [21,23]. However, these authors did not use electron microscopy techniques to study the glycocalyx layer, making comparisons to our study difficult. Instead, these studies collectively provide support for a decreased glycocalyx being associated with increased microbial adherence to the epithelium and increased permeability, which could influence IBS pathogenesis and symptom severity.

In this study, we confirm humoral immunity as a specific molecular signature in IBS-D. Notably, 58% of the top 50 genes with positive enrichment scores are related to IgG structure, specifically to variable regions, which, through somatic hypermutation, diversify the antibody repertoire to more specifically bind to antigens. Notably, IBS-D showed higher lambda chain transcription, which is the predominant chain in mucosal antibodies [58]. Regardless of these molecular events needing further insight, our observations suggest the increase in intestinal IgG is secondary to persistent mucosal antigenic exposure.

Despite gastrointestinal infection being recognized as a risk factor in IBS and several studies identifying intestinal dysbiosis, the contribution of microbial factors in the onset of IBS remains unclear. Lipopolysaccharide (LPS) and flagellin have been associated with IBS [15,59]. More specifically, LPS has been linked with intestinal inflammation and visceral hypersensitivity, and TLR4 expression is also increased in the jejunum of IBS patients compared to healthy controls [60,61]. Here, despite not focusing on the type of antigens promoting humoral responses, we identify higher intestinal IgG transcripts and protein content in IBS-D, specifically of IgG2, with respect to HC. IgG2 plays an important defensive barrier role against LPS encapsulated microbes [62,63], suggesting that increased IgG2 production and secretion could respond to microbial invasion due to epithelial barrier breach. Additionally, *Brachyspira*, a gram-negative bacterium, is present in a subgroup of IBS patients, together with the invasion of gram-positive bacteria in colonic crypts [41], providing a rationale for performing broader surveillance of microbial components to detect translocation of microbial agents that do not fall into the gram-negative criteria. To reduce microbial invasion, mast cells could also play a role in acquired immunity, as they express Fc receptors for IgG. Here we observed that the number of mast cells correlates with plasma cell activation, and mast cells were also identified in proximity to plasma cells [13]. These observations support further studies to more precisely define the contribution of mast cells to mucosal defence in IBS-D. Additionally, within the genes with positive enrichment scores, we identified lipocalin-2, an antimicrobial innate immune protein expressed by epithelial cells and neutrophils [64], suggesting a range of microbial components to elicit innate defensive responses in IBS-D.

At the ultrastructural level, we show a differential activation of plasma cells by assessing the number of mitochondria and the proportion of RER in PC cytoplasm [35,65]. Indeed, the association between these two surrogate cell activation markers suggests that increased energy metabolism underlies antibody production [65]. These antibody-producing cells are also closer to nerve fibres in IBS-D compared to healthy controls, a neuro-immune interaction that presumably maintains plasma cell activation, differentiation, survival, and secretion [66,67,68,69]. This proximity could represent an epiphenomenon related to mucosal inflammation, as described in IBD [70]; however, the degree of inflammation is not comparable between these two entities. We did not find strong associations between plasma cell proximity to nerves and clinical manifestations, suggesting this interaction may have local relevance in maintaining defensive responses epithelial barrier is altered. In fact, reduced intestinal glycocalyx strongly correlates with plasma cell proximity to nerves, supporting a potential epithelial-neuro-immune axis as relevant to mucosal defence in IBS-D.

In IBS, the communication between plasma cells and nerves in the intestinal mucosa has not been previously addressed, but several mediators could be implicated. Of relevance for IBS are the neuropeptides nerve growth factor (NGF), substance P (SP), and vasoactive intestinal peptide (VIP), all differentially expressed in the intestinal mucosa of IBS and involved in motility, visceral sensitivity, and barrier mechanisms [46,66,67,68,71,72]. Notably, NGF promotes the growth and survival of both neurons and plasma cells [73], and SP and VIP increase immunoglobulin synthesis [66,67]. Under inflammatory conditions, mucosal plasma cells also express urocortin-1, which exerts an excitatory effect on enteric nerves [74,75]. However, the mechanisms of plasma cell-nerve communication and its implication in IBS-D deserve further investigation. Unfortunately, we could not quantify IgG in faecal samples from the same participants in which the ultrastructural analysis was performed. It would be interesting to study whether mucosal plasma cells modulate nerve cell activity, as the increase in luminal IgG correlates with the frequency and intensity of abdominal pain.

We acknowledge several limitations of this study. The fixation process for preserving the full thickness of the glycocalyx could be improved, and there is an ongoing debate regarding how to optimally fixate the glycocalyx, with no consensus yet reached on the matter [57,76]. Studies on glycocalyx thickness in the small intestine are limited to animal studies, where the results are not comparable to ours [37]. However, we carried out the same fixation process in all samples and in a blinded fashion, thus allowing a fair comparison. Our fixation also provides a quality that can display finer details in the glycocalyx meshwork, demonstrating good methodology with high quality, as previously validated [22], and even more advantageous compared to fixation of alveolar tissue [76]. Also, the low number of available samples for some analyses could limit the interpretation of the results; therefore, additional studies are needed to validate our findings. Assessing the correlation between biological and clinical variables does not reveal direction or causality; however, our results strongly support a potential epithelial-neuro-immune link as a plausible mechanism to contribute to IBS-D pathophysiology. The findings of this study could help to define new strategies to search for biomarkers related to mucus biology and plasma cell activity, and could also provide the rationale for therapeutic interventions. Thus, strengthening the glycocalyx could be critical in conditions characterised by increased gut permeability and low-grade mucosal inflammation. There could be a potential for future nutritional interventions in rescuing glycocalyx integrity or complementing it by using, for example, mucoprotectants, milk oligosaccharides, or non-digestible carbohydrates [77,78,79].

## 5. Conclusions

Our results highlight the importance of intestinal barrier integrity and humoral activity in IBS-D and its relationship to major symptoms. These findings provide a rationale for further investigating the type of antigens associated with such defensive response and the potential role of the epithelial-neuro-immune axis in IBS-D pathophysiology.

## Figures and Tables

**Figure 1 cells-11-02046-f001:**
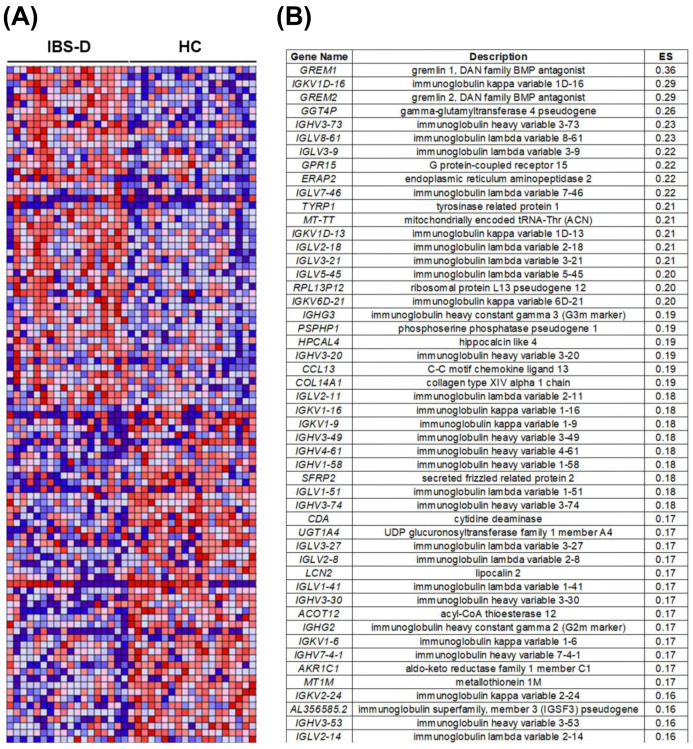
Gene markers for the IBS-D versus HC comparison identified by GSEA. (**A**) Representation of the heat map for the top 50 positive and top 50 negative enriched genes correlated to IBS-D, indicating gene set enrichment at the top and the bottom of the ranked list, respectively. Expression values for the top-ranked genes are represented as colours, where the range of colours (red, pink, light blue, dark blue) shows the range of expression values (high, moderate, low, lowest). (**B**) Top 50 ranked gene list in IBS-D compared to HC. Degree of overrepresentation is indicated with ES (enrichment score); positive ES indicates gene set enrichment at the top of the ranked list. HC: Healthy control; IBS-D: diarrhoea-predominant Irritable Bowel Syndrome.

**Figure 2 cells-11-02046-f002:**
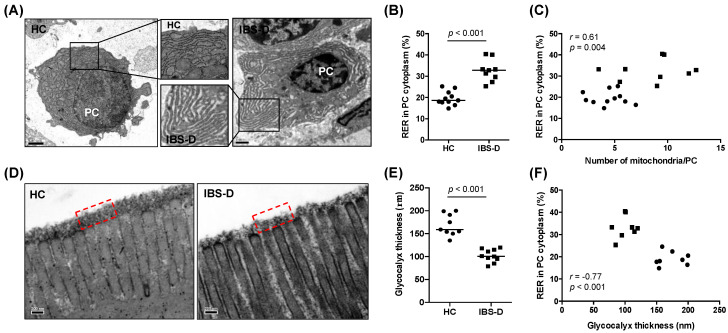
Plasma cell ultrastructure and glycocalyx thickness in the jejunal mucosa of HC subjects and IBS−D patients. (**A**) Representative transmission electron microscopy images of plasma cells from HC (left panel) and IBS−D (right panel) groups. Inserts show the morphology of the RER in each group. Bar represents 1 µm. (**B**) Quantification of the percentage of RER area in the cytoplasm (%RER) in plasma cells in HC (n = 11) and IBS−D (n = 9) groups. (**C**) Correlation assessment between the %RER and the number of mitochondria per plasma cell (pooled participants, n = 20). (**D**) Representative transmission electron microscopy images of the glycocalyx overlying the microvilli from HC (left panel) and IBS-D (right panel) groups. Bar represents 200 nm; dashed square height indicates 260 nm. (**E**) Quantification of the glycocalyx thickness in HC (n = 9) and IBS−D (n = 10) groups. (**F**) Correlation assessment between the %RER and the glycocalyx thickness (pooled participants, n = 16). IBS−D: diarrhoea−predominant irritable bowel syndrome; HC: healthy control; PC: plasma cell. Mann−Whitney U test was used for comparisons between groups and Pearson’s correlation coefficient analysis for correlations between the parameters. Differences were considered significant when *p* ≤ 0.05.

**Figure 3 cells-11-02046-f003:**
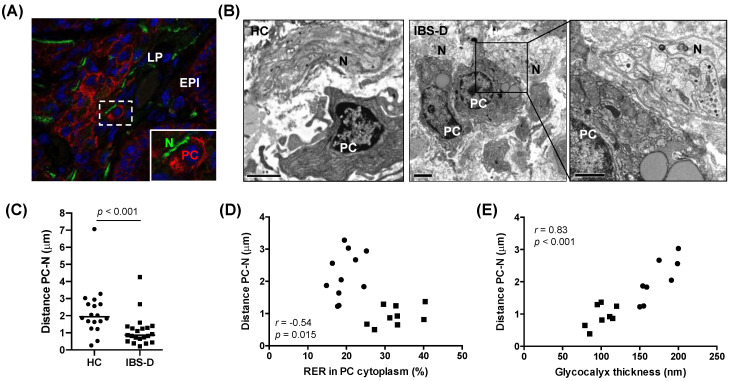
Plasma cells and nerve fibres in the jejunal mucosa of HC subjects and IBS−D patients. (**A**) Representative immunofluorescence staining of the plasma cell terminal differentiation marker CD138 (red), the neuronal marker PGP9.5 (green), and cell nuclei (blue, DAPI) from an IBS−D patient. This neuro−immune interaction, marked by a dashed square (white), is highlighted via an enlarged insert. (**B**) Representative electron microscopy images showing an association between plasma cells and nerves in HC (left panel) and IBS-D (right panel). (**C**) Quantification of the distance (µm) from plasma cell membrane to nerve fibres in HC (n = 18) and IBS-D patients (n = 21). (**D**) Correlation between plasma cell−nerve fibres distance and the area of rough endoplasmic reticulum (%RER) in plasma cells (pooled participants, n = 20). (**E**) Correlation between plasma cell−nerve fibres distance and the glycocalyx thickness (pooled participants, n = 16). IBS-D: diarrhoea−predominant irritable bowel syndrome; HC: healthy control; PC: plasma cells; N: nerve fibres; EPI: epithelium; LP: lamina propria. Mann−Whitney U test was used for comparisons between groups and Spearman’s correlation analysis for correlations between the parameters. Differences were considered significant when *p* ≤ 0.05.

**Table 1 cells-11-02046-t001:** Demographic, clinical, and histological characteristics.

	HC	IBS-D	
	(n = 63)	(n = 71)	*p*
**Sex**, F:M	29:34	50:21	**0.005**
**Age**, years (range)	25 (18–54)	36 (18–64)	**<0.0001**
**Intensity of abdominal pain**, score	-	44.5 (2–100)	-
**Frequency of abdominal pain**, number of days	-	5 (1–10)	-
**Bowel movements**, number/day	1.5 (0.5–2.5)	3.0 (0.5–12)	**<0.0001**
**Stool form**, Bristol score	3.5 (2–4.5)	5.5 (3–7)	**<0.0001**
**Dyspepsia** (Yes/No)	-	26/41	-
**IBS-SSS**	-	268.5 ± 91.9	-
***Dyspeptic***	-	*291.4 ± 95.2*	*0.019 **
***Non-dyspeptic***	-	*236.1 ± 77.4*	
**Holmes-Rahe scale**, score	88 (0–357)	111 (0–889)	0.145
**Cohen scale**, score	14 (4–37)	24 (1–41)	**<0.0001**
**Beck’s Depression Inventory**, score	0 (0–22)	8 (1–31)	**<0.0001**
**CD117^+^** (cells/hpf)	20.1 (2.0–59)	25.4 (5.6–59)	0.185
**CD3^+^** (cells/100 enterocytes)	17 (7.0–40)	16 (1.6–37)	0.746
**Eosin^+^** (cells/hpf)	1.5 (0–5.6)	1.7 (0–18)	0.213

Results are expressed as median (minimum-maximum values) or mean ± SD; *p* values considered significant are shown in bold. ***** Differences in IBS-SSS between dyspeptic and non-dyspeptic IBS-D patients. F: female; HC: healthy control; Hpf: high power field; IBS-D: diarrhoea-predominant Irritable Bowel Syndrome; IBS-SSS: IBS severity score system; M: male.

**Table 2 cells-11-02046-t002:** Representative selection of enriched Reactome pathways in IBS-D.

Reactome Pathway	*NES*	*p*	*FDR*
PTEN regulation	1.88	0.027	0.14
Regulation of apoptosis	1.73	0.010	0.14
Interleukin 1 signalling	1.90	0.008	0.15
Activation of NF-KB in B cells	1.91	0.004	0.15
Cellular responses to stress	1.75	0.006	0.15
Protein ubiquitination	1.78	0.033	0.15
Downstream signalling events of B cell receptor	1.97	0.002	0.16
Dectin 1 mediated non canonical NF-KB signalling	1.69	0.032	0.17
TNFR2 non-canonical NF-KB pathway	1.65	0.025	0.18
Cell extracellular matrix interactions	1.60	0.073	0.19
Interleukin 1 family signalling	1.59	0.047	0.19
Signalling by the B cell receptor	1.58	0.042	0.19
Antigen processing cross presentation	1.63	0.060	0.19
Programmed cell death	2.04	0.0001	0.20
Regulation of actin dynamics for phagocytic cup formation	1.56	0.034	0.21
Antigen activates B cell receptor leading to generation of second messengers	1.55	0.019	0.21
FCERI-mediated NF-KB activation	1.54	0.049	0.22
Cross presentation of soluble exogenous antigens endosomes	1.54	0.071	0.22
Complement cascade	1.52	0.021	0.22
Neutrophil degranulation	1.49	0.058	0.24
Class I MHC mediated antigen processing presentation	1.50	0.033	0.24
Gap junction assembly	1.50	0.063	0.24
Gap junction trafficking and regulation	1.50	0.062	0.24
Tight junction interactions	1.48	0.059	0.24

Gene sets from the IBS-D phenotype were compared to curated data sets from Reactome. Significance of this enrichment and degree of overrepresentation is indicated with FDR and NES (normalized enrichment score) values, respectively.

**Table 3 cells-11-02046-t003:** Representative selection of enriched immunological signatures in IBS-D.

Immunologic Signature	*NES*	*p*	*FDR*
**Plasma cell activity**			
Down-regulated genes in naive B cells vs. plasma cells from bone marrow and blood	1.86	0.01	0.20
Down-regulated genes in memory IgM B cells vs. plasma cells from bone marrow and blood	1.67	0.04	0.23
Down-regulated genes in B lymphocytes from peritoneal cavity vs. spleen	1.63	0.07	0.23
Up-regulated genes in plasma cells vs. memory B cells	1.75	0.01	0.23
**Germinal centre activity**			
Upregulated genes in IgD-peripheral blood B cells vs. dark zone germinal centre B cells	1.87	0.01	0.22
Down-regulated genes in naive B cell vs. dark zone germinal centre B cells	1.83	0.01	0.23
Down-regulated genes in IgD+ peripheral blood B cells vs. dark zone germinal centre B cells	1.66	0.03	0.23
Down-regulated genes in pre-germinal centre B cells vs. dark zone germinal centre B cells	1.81	0.01	0.23
Down-regulated genes in naive follicular B cells vs. early germinal centre B cells	1.63	0.06	0.24
Upregulated genes in germinal centre B cells vs. naive B cells	1.61	0.04	0.24
**Predominance of B cells**			
Down-regulated genes in B cells vs. plasmacytoid dendritic cells	1.68	0.03	0.22
Down-regulated genes in Neutrophils vs. B cells	1.70	0.04	0.22
Down-regulated genes in naive B cells vs. day 0 monocytes	1.61	0.07	0.24
Down-regulated genes in naive B cells vs. plasma cells	1.58	0.02	0.25
Up-regulated genes in B cells vs. monocyte macrophages	1.60	0.03	0.25
Down-regulated genes in naive CD4 T cells vs. naive B cells	1.61	0.06	0.25

Gene sets from the IBS-D phenotype were compared to immunological signatures in public repositories. Significance of this enrichment and degree of overrepresentation is indicated with FDR and NES (normalized enrichment score), respectively. Positive ES indicates gene set enrichment at the top of the ranked list; a negative ES indicates gene set enrichment at the bottom of the ranked list.

**Table 4 cells-11-02046-t004:** Concentration of Ig in stool from HC and IBS-D participants.

Ig (*ng/mg Protein*)	HC (n = 17)	IBS-D (n = 19)	*p*
sIgA	11.55 × 10^3^ (1.34–71.30)	26.78 × 10^3^ (4.02–221)	0.099
IgM	62.54 (3.44–1074)	109.1 (4.75–1854)	0.128
IgG	3.68 (0.61–35.34)	11.38 (1.5–45.82)	**0.041**
IgG1	0.15 (0.003–0.82)	0.07 (0.002–96)	0.274
IgG2	0.40 ± 0.29	0.71 ± 0.53	**0.039**
IgG3	0.91 (0.19–1.83)	1.05 (0.28–3.65)	0.079
IgG4	0.028 ± 0.004	0.021 ± 0.004	0.184
IgE	0.090 (0–0.472)	0.159 (0.012–2.590)	0.104

Results are expressed as [ng Ig/mg protein], except for sIgA which is [µg Ig/mg protein]. Ig: immunoglobulin; HC: healthy control; IBS-D: diarrhoea-predominant irritable bowel syndrome.

**Table 5 cells-11-02046-t005:** Correlation analysis between clinical, histological, and biological variables.

	Glycocalyx Thickness			%RER in PC Cytoplasm			PC-N Distance		
*Variables*	*r_s_*	*p*	FDR	*r_s_*	*p*	FDR	*r_s_*	*p*	FDR
**Age**	−0.23	0.336	0.56	0.2	0.397	0.662	0.13	0.443	0.633
**Intensity of abdominal pain**	0.34	0.365	0.521	0.55	0.196	0.57	0.18	0.45	0.563
**Frequency of abdominal pain**	0.69	**0.047**	0.94	0.52	0.196	0.49	0.22	0.358	0.716
**Number of bowel movements**	−0.73	**0.007**	**0.035**	0.64	**0.029**	**0.145**	−0.18	0.365	0.608
**Stool form**	−0.66	**0.007**	**0.023**	0.57	**0.024**	**0.24**	−0.27	0.123	0.41
**Dyspepsia**	−0.21	0.834	1	−0.13	0.857	0.857	0.02	0.937	0.937
**IBS-SSS**	−0.03	0.938	0.938	0.24	0.582	0.831	0.1	0.678	0.753
**-*Dyspeptic***	0.02	0.989	-	0.54	0.297	-	−0.04	0.91	-
**-*Non-dyspeptic***	- ^Ŧ^	- ^Ŧ^	-	- ^Ŧ^	- ^Ŧ^	-	0.2	0.714	-
**Holmes-Rage scale**	0.04	0.887	0.986	−0.1	0.671	0.746	0.17	0.318	0.795
**Cohen scale**	−0.52	**0.035**	**0.088**	0.14	0.589	0.736	−0.35	**0.032**	0.32
**Beck’s Depression Inventory**	−0.73	**<0.001**	**0.006**	0.28	0.248	0.496	−0.3	**0.066**	0.33
**CD117^+^ cells**	−0.59	**0.008**	**0.024**	0.68	**0.001**	**0.003**	−0.21	0.209	0.209
**CD3^+^ cells**	−0.51	**0.027**	**0.041**	0.27	0.245	0.368	0.27	0.106	0.159
**Eosin^+^ cells**	−0.37	0.125	0.125	0.27	0.253	0.253	−0.33	**0.047**	**0.141**

Glycocalyx thickness (HC n = 5–9; IBS-D n = 8–10); %RER in PC cytoplasm (HC n = 5–11; IBS-D n = 7–9); The analysis of abdominal pain and frequency, abdominal distension, IBS-SSS and dyspepsia only considers the IBS-D group. PC-N distance (HC n = 10–18; IBS-D n = 19–21). HC: healthy control; IBS-D: diarrhoea-predominant irritable bowel syndrome; IBS-SSS, IBS severity score system; PC: plasma cells; N: nerve fibres. *r_s_* = Spearman correlation analysis. ^Ŧ^ Too few participants for analysis.

## Data Availability

All data generated during the study are available from the corresponding authors upon request.

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
