# Peer review of "Mucosal Plasma Cell Activation and Proximity to Nerve Fibres Are Associated with Glycocalyx Reduction in Diarrhoea-Predominant Irritable Bowel Syndrome: Jejunal Barrier Alterations Underlying Clinical Manifestations"

_cells, 2022, doi:10.3390/cells11132046_

Round 1

Reviewer 1 Report

This is a complex and interesting paper dealing with the difficult issue of investigating the mucosal barrier alterations and immune alterations in patients with IBS, diarrhea subtype. The main findings are reduction of the glycocalyx, signs of an activated humoral immune response, and proximity of plasma cells to nerve fibers in IBS-D patients. The paper definitely deserves publication.

However, I would propose some minor issues that will further improve the scientific value:

- please clarify how the patients were chosen for this study (special time period or random).

- the author should clearly where in the mucosa the samples for TEM are taken (basal layer of the mucosa, mid part, villi)? Is there basal plasmacytosis like in IBD patients?

- the authors investigate the expression of MUC 17, but not MUC13 which is as well a part of the glycocalyx. Why not? (not mandatory).

- the authors describe no differences in mucosal mast cells and eosinophils between IBS patients and HC. Since the paper is dealing with the humoral immune response it is recommended to include the distribution of B-cells (CD20) and plasma cells, (if possible IgG-positive plasma cells).

- the authors previously reported an increased mast cell activity. Is there a correlation between the activity in mast cells and B cells/ plasma cells? It should be shortly discussed.

- the authors show the nicely close proximity of plasma cells and nerve fibers (fig. 3 A), but they could not show any signs of activated nerve cells. Is there any indication for this in the transcriptome data? Will this be a topic of further investigation? The authors discuss that plasma cells may modulate nerve cell activity. Why could it not be the reverse way?

- In the TEM pictures (fig. 3 B) the nerve fibers look very different in HC and IBS-D. This should be thoroughly described as like in the plasma cells.

Reviewer 2 Report

The work by Pardo-Camacho et al describes an interesting link between plasma cells and its proximity to nerve fibers in affecting glycocalyx thickness. The manuscript is well presented and the following minor modifications are recommended

1. The authors describe here jejunal region, have the authors looked into  any other parts of the intestine to see if a similar mechanism occurs? please speculate

2. With recent work on the importance of serotonergic pathway in IBS. Does this pathway and mediators have any role in the plasma cell activation/ glycocalyx? discuss

3. It will be good if amount of mucus is shown in the patient jejunal samples by a staining such as PAS? Is this feasible or has this already been done?
